# Adhesion and Stability of Nanocellulose Coatings on Flat Polymer Films and Textiles

**DOI:** 10.3390/molecules25143238

**Published:** 2020-07-16

**Authors:** Raha Saremi, Nikolay Borodinov, Amine Mohamed Laradji, Suraj Sharma, Igor Luzinov, Sergiy Minko

**Affiliations:** 1Nanostructured Materials Laboratory, University of Georgia, Athens, GA 30602, USA; raha@uga.edu (R.S.); alaradji@wustl.edu (A.M.L.); ssharma@uga.edu (S.S.); 2Department of Textiles, Merchandising and Interiors, the University of Georgia, Athens, GA 30602, USA; 3Department of Materials Science and Engineering, Clemson University, Clemson, SC 29634, USA; nikolab@g.clemson.edu (N.B.); luzinov@clemson.edu (I.L.)

**Keywords:** nanocellulose, polymer, coating, textile, adhesion

## Abstract

Renewable nanocellulose materials received increased attention owing to their small dimensions, high specific surface area, high mechanical characteristics, biocompatibility, and compostability. Nanocellulose coatings are among many interesting applications of these materials to functionalize different by composition and structure surfaces, including plastics, polymer coatings, and textiles with broader applications from food packaging to smart textiles. Variations in porosity and thickness of nanocellulose coatings are used to adjust a load of functional molecules and particles into the coatings, their permeability, and filtration properties. Mechanical stability of nanocellulose coatings in a wet and dry state are critical characteristics for many applications. In this work, nanofibrillated and nanocrystalline cellulose coatings deposited on the surface of polymer films and textiles made of cellulose, polyester, and nylon are studied using atomic force microscopy, ellipsometry, and T-peel adhesion tests. Methods to improve coatings’ adhesion and stability using physical and chemical cross-linking with added polymers and polycarboxylic acids are analyzed in this study. The paper reports on the effect of the substrate structure and ability of nanocellulose particles to intercalate into the substrate on the coating adhesion.

## 1. Introduction

Cellulose is the most abundant [1,2], renewable, biodegradable, and environmentally friendly organic material found in nature with great potential for the development of new applications with minimal health, environmental, or safety concerns [3]. Plant cell walls are composed of assembled cellulosic fibrils that are stabilized by intra- and interchain hydrogen bonds and van der Waals forces [4]. The fibrils are semicrystalline cellulose with 50–75% crystalline regions [5]. The fibrils can be separated by mechanical [2,6,7,8,9,10], chemical [11,12,13], or a combination of both treatments [14] to make cellulose nanoparticles in the form of nanofibers (nanofibrillated cellulose, NFC) or whiskers (nanocrystalline cellulose, NCC) [15,16] forming hydrogels in water. These nanoparticles, owing to their dimensions, shape, and high mechanical characteristics, attracted great interest in the engineering of nanostructured materials [14,17,18,19,20,21]. Nanocellulose-based materials, including coatings, were explored for many applications, such as packaging films [22,23,24,25,26], engineered composites [27], adsorbents [28], materials for health care [29], cosmetics [30], thermal insulation [31], paper [32], and filtration [33,34].

For time immemorial, one of the traditional application of the cellulosic material is the textile industry. Cotton, linen, hemp, and many other plant fibers are the major feedstock for the most demanded and comfortable cloth. In recent decades, there is a clear disposition for the shift from conventional clothing to smart textiles that integrate emerging technologies, such as communication devices, flexible electronics, and sensors [35]. Importantly, strong environmental and societal concerns demand a shift to the development of renewable and compostable materials along with sustainable technologies with minimal negative environmental impact [36]. A combination of wearable clothing systems and sustainability is advancing innovation in the traditional areas of textile manufacturing by endowing textiles with functional properties to address current health, safety, and environmental concerns associated with the textile industry.

Nanoscale dimensions and large specific surface of NFC and NCC allow them to intercalate into hierarchically organized fibrous structures, such as woven, knits, nonwoven, and composite textiles. Hence, nanocellulose materials can deliver functional molecules or particles bearing functionality covalently bonded or physically entrapped (caged) into a nanocellulose particle network. The functionalized NFC and NCC network is subsequently anchored to the textile surfaces via hydrogen/covalent bonds and physical interlocking. This method of functionalization of natural and synthetic fibers and fabrics is an environmentally sound approach without compromising the compostability and biocompatibility of the compostable textiles. For example, we have recently demonstrated that dyeing of textiles using NFC particles conjugated with commonly used reactive dyes can decrease the use of water, salts, and alkali by one order of magnitude with no change of the textile performance such as colorfastness referred to as conventional textile dyeing technology [37,38,39].

The key aspects of NFC and NCC textile coatings are adhesion and mechanical stability during dry (wearing due to abrasion) and wet (laundry) conditions. Cellulose has a natural self-adhesive characteristic, which relies primarily on interchain hydrogen bonding between hydroxyl groups of the adjacent cellulose chains and between cellulose chains and polymers of the textile, entanglements (specifically for NFC), and interlocking through the entanglement and intercalation into the fabric structures [40,41]. Many pretreatment methods such as plasma, ozone, and exposure to alkali solutions were successfully used for the materials incapable of the formation of hydrogen bonds with nanocellulose. For example, the treatment of polypropylene with ozone resulted in an improved adhesion to NCC owing to the hydrogen bonds with the oxidized surface of polypropylene bearing hydroxyl, carboxylic, and other oxygen-containing functional groups [42]. Nevertheless, strongly hydrogen-bonded nanocellulose materials swell in an aqueous environment [43,44]. The swelling can cause film degradation and loss of functional properties bound to the nanocellulose coatings. Therefore, it is important to understand the effect of nanocellulose swelling on the coating stability and develop methods for mitigating this problem.

This paper reports on a systematic study of adhesion and adhesion resistance to swelling (in water) of NFC and NCC thin film coatings on the surface of polymer films and fabrics made of cellulose (CL), cotton, poly(ethylene terephthalate) (PET), and nylon 6,6 (PA 6,6). We studied several methods for improvement of the adhesion and coating’s stability, such as the use of a cationic polyelectrolyte poly(ethylene imine) (PEI), functional copolymers, and covalent cross-linking to elucidate major mechanisms for the improvement of the stability of nanocellulose coatings via combinations of adhesive and cohesive properties of the coatings. PEI is added to enforce the physical network of nanocellulose particles and nanocellulose-polymer substrate (films and textiles) interfaces via strong hydrogen bonds between primary and secondary amino groups of PEI and hydroxyl, amide, and ester groups of nanocellulose and textile materials (PET and nylon). A functional polymer—a copolymer of glycidyl methacrylate (GMA) and oligo(ethylene glycol) methacrylate (OEGMA) (P(GMA-OEGMA))—was selected and synthesized based on the compatibility with nanocellulose hydrogels, the ability to form hydrogen bonds with cellulose, and functional epoxy groups to cross-link the polymer and form a network for reinforcing of the coating. Alternatively, a commonly used cellulose-crosslinking method with polycarboxylic acids was applied to probe the effect of cross-linking on the coating stability.

## 2. Results and Discussion

### 2.1. Fabrication of Uniform Nanocellulose Coatings on the Surface of Polymeric Materials

The formation of nanocellulose coatings on the surface of fabrics is affected by the infiltration of nanocellulose hydrogels into a complex structure of the fabric. The permeation dynamics of hydrogels depends on the fabric density, structure of the yarn, interfacial tension, and rheological properties of the hydrogel. Many of these complications can be eluded using single filament fibers for coating, where the film formation is only limited by the wetting thermodynamics and rheology of the hydrogel. For low nanocellulose concentrations (<1%), when the hydrogel viscosity is low, it spreads over the fiber surface and forms an enclosed nanocellulose coating upon evaporation of water, as can be observed from the scanning electron microscopy (SEM) images of the coated polyester, cotton, and nylon single fibers (Figure 1). The image of a nylon fiber (Figure 1c) exhibits a peeled off NFC film at the edge of the cut fiber surface visualizing the coating film morphology and thickness. The peeled fraction of the coating corroborates a uniform layer of NFC around the fiber surface.

The results of the experiments with single filament fibers show the formation of uniform smooth coatings over the fiber surface (Figure 1a,b). This uniform coating of NFC justifies the use of model flat substrates (e.g., polymer films) to probe morphology, adhesive behavior, and stability of nanocellulose coatings on the surfaces of different polymeric materials to monitor the coating structure and changes upon different treatment methods. Figure 2 exhibits differences in the surface morphology of NFC and NCC coatings on the Si-wafers. NFC coatings show a higher roughness owing to the higher particle size polydispersity in contrast to smoother and more uniform NCC coatings.

In this work, we used plane polymer films made of CL, PET, and PA 6,6 polymer solutions deposited on the surface of polished Si-wafers to minimize possible effects of the surface roughness of the substrate on the film formation. We prevented possible instabilities that could originate from a poor polymer-Si-wafer adhesion via the strengthening of the polymer-Si-wafer interactions by pretreatment of the Si-wafers with PEI polycations prior to deposition of the CL, PET, and PA 6,6 polymer films. Thickness and surface roughness of the polymer films were estimated with ellipsometry and atomic force microscopy (AFM) (Figure 3). The root mean square (RMS) roughness of the films did not exceed 7 nm, with the highest roughness observed for the cellulose films.

NFC and NCC films were deposited (spin-coated) on the surface of the polymer-coated Si-wafers using several different protocols (Appendix A). (i) Protocol 1: NFC and NCC aqueous dispersions were deposited on the CL, PET, and PA 6,6 coated Si-wafers; (ii) Protocol 2: NFC and NCC aqueous dispersions were deposited on the PEI pretreated CL, PET, and PA 6,6 coated Si-wafers; (iii) Protocol 3: NFC and NCC aqueous dispersions were mixed with PEI, and spin-coated on the CL, PET and PA 6,6 coated Si-wafers; and (iv) Protocol 4: NFC and NCC aqueous dispersions were mixed with P(GMA-OEGMA) copolymer, spin-coated on the CL, PET, and PA 6,6 coated Si-wafers. In all cases, the nanocellulose coatings were annealed after the deposition at 120 °C for 1 h.

The NFC and NCC coatings were prepared first using Protocol 1. We discovered a poor coverage of the PET and PA 6,6 surface with the nanocellulose materials. Then, we applied PEI pretreatment of all polymer substrates in Protocol 2 to improve wetting and coverage with nanocellulose. NFC- and NCC-coated samples from Protocol 2 are labeled as PEI-NFC and PEI-NCC, respectively.

Alternatively, we applied two different protocols to improve the adhesive and cohesive properties of nanocellulose coatings. According to Protocol 3, we mixed NFC and NCC hydrogels with PEI in solutions prior to the deposition on the polymer surfaces. Obtained by Protocol 3, NFC- and NCC-coated samples are labeled as NFC+PEI and NCC+PEI, respectively. Protocol 4 was used to mix NFC and NCC hydrogels with P(GMA-OEGMA) copolymer solutions; the samples were labeled as NFC+CP and NCC+CP, respectively.

To summarize the nanocellulose coatings preparation, the resulting films are multilayered structures constituted of the Si-wafer substrate, a native SiO_2_ layer (0.5–1 nm), PEI adsorbed layer (typically 0.2–0.5 nm thick), polymer coating (CL, PET, or PA 6,6, typically 50–180 nm thick), and a nanocellulose (NFC or NCC) top layer with or without mixing with PEI or the copolymer. In some samples, the polymer layers are pre-coated with PEI (Protocol 2) before applying NC coatings. Representative 3D-plots for the layered structures obtained with imaging ellipsometry demonstrate uniform layered structure across the multicomponent coatings (Figure 4).

### 2.2. Mechanisms of the Nanocellulose Coating Degradation in a Wet State

In aqueous solution, nanocellulose coatings become swollen owing to the strong hydrogen bonding of water molecules and cellulose [44]. The developed osmotic pressure, in combination with share forces, can cause complete defoliation of the coatings from the substrate surface or partial delamination. The prevalence of one of the two mechanisms of degradation is defined by the balance of adhesive and cohesive interactions in the film. The complete or very large depletion of the film materials is likely associated with adhesive failure, while fractional losses or partial delamination of the coating film are caused by cohesive failure.

The stability of the deposited NFC and NCC films in an aqueous environment was estimated with a simple test. The coated samples were exposed to 50 °C aqueous solution at stirring for 1 h. Comparing the AFM images of the film before and after exposure to the aqueous medium in most cases did not reveal changes in the film morphology (Figure 5). For these nanocellulose coatings, we monitored changes in average film thickness. Only in the case of very poor adhesion, as for untreated PET and some PA 6,6 substrates, the AFM images demonstrate a low surface coverage by NFC and NCC, respectively (Figure 6 and Figure 7). For these coatings, we monitored the surface coverage using an AFM “flooding analysis”—a statistical method where the polymer coating layer is set as a threshold and the surface areas of all other structures above the threshold are added to estimate the overall coverage with the nanocellulose material.

Changes in film thickness of the NFC and NCC coatings in all cases, with exceptions of untreated PET substrates, after rinsing in water, are reported in Table 1 as a percent (%) of the detached coating materials. For untreated PET substrates, the results report changes of the surface coverage using the flooding method. The coating thickness prepared by a spin-coating method depends on the rheological characteristics of NFC and NCC hydrogels and wetting of the polymer films. The rheological properties of the hydrogels depend on the concentration and presence of additives. Consequently, we analyze the relative changes in film thickness prepared using different modification methods.

The analysis of the experimental data shows that the most common mechanism of coating degradation is partial delamination. Only for the PET substrate, we observed almost a complete adhesive detachment of the nanocellulose. The result shows that the nanocellulose coating has the lowest adhesion to the PET surface and the strongest interaction with the PA 6,6 surface among the synthetic polymers. NCC coatings demonstrate a higher adhesion to different CL substrates than NFC coatings. It is likely owing to the denser packed NCC particles in the coating in contrast to NFC fibers, and hence a lower swelling of the coating. The addition of PEI and P(GMA-OEGMA) improves the stability of the coatings. The latter effect is likely owing to the switching from the adhesive defoliation mechanism to the partial delamination of the film.

Notably, the film is much more stable on PET and PA 6,6 substrates when the coating is mixed with P(GMA-OEGMA). We may speculate that the major strengthening contribution of the copolymer is in the improvement of the cohesive properties of the film. The much greater thickness of the mixed films supports this conclusion.

We may speculate about the following mechanism of the improvement of the stability of the coating in an aqueous environment. Cationic anchoring polymers bearing amino-functional groups are used to treat different substrates for improvement in their interaction with cellulose [45,46,47,48,49,50,51,52,53]. Cationic polyelectrolytes interact with cellulose coatings through an electrostatic, donor-acceptor type of interactions, and hydrogen bond formation [45]. Nanocellulose, cotton, and polyester fibers are negatively charged in an aqueous environment, whereas nylon possesses amphoteric properties. In all cases, swollen in water, nanocellulose materials and polymers experience repulsive electrostatic forces. These repulsive interactions could be compensated by surface modification of the interacting materials with polycations such as PEI [52]. However, an excess of PEI will result in an overcharge of the surfaces, and repulsion between negatively charged materials will be replaced by repulsion among positively charged materials in water. PEI and other polyelectrolytes may also enhance swelling of the coatings in water. The results show no benefits of the use of PEI for the treatment of cotton and nylon fabrics; however, the interaction with PET is slightly improved. The latter is explained by poor hydrogen bonding between PET and nanocelluloses, which can be improved owing to the PEI-PET interactions.

P(GMA-OEGMA) copolymer bears ethylene oxide and epoxy functional groups. These two types of functional groups provide a combination of strong hydrogen bonds and covalent cross-linking. The covalent cross-linking mechanism involves the opening of epoxy rings and the formation of covalent bonds between epoxy groups of P(GMA-OEGMA) [54]. Reactivity of cellulosic -OH with epoxy-groups is not high enough to provide substantial effect for the cross-linking involving nanocellulose [55]. However, surface carboxylic functional groups that may be present because of oxidative degradation in the process of the production of nanocellulose could interact with epoxy groups and form covalent cross-links. The nanocellulose and P(GMA-OEGMA) copolymer blends upon drying, and thermal annealing will form an interpenetrated network owing to the covalently cross-linked polymer and physical cross-links via hydrogen bonds of nanocellulose materials. These two interpenetrating networks are also co-cross-linked via some fraction of carboxylic groups on the surface of cellulose. Epoxy groups secure good adhesion to various polar substrates. The experiments show that the presence of the copolymer improves adhesion to PET and nylon surfaces. In all experiments, we observe the obvious improvement of the nanocellulose coating stability in the presence of the copolymer.

### 2.3. Adhesive Behavior of Nanocellulose Coatings

Mechanical stability of the nanocellulose coating in the dry state is another important property for practical applications. Upon mechanical forces, the coating could be peeled off the polymer surface (adhesive failure) or partially delimited (cohesive failure). Hence, the coating performance can be analyzed using similar concepts of the degradation mechanisms, as was discussed for the aqueous environment. For the experiments, we used T-peel tests. Two identical materials were adhered using NFC or NCC hydrogels sandwiched between the materials, followed by drying and annealing the samples. These tests were performed in two series of experiments. In the first series, NFC and NCC were used to bind two identical samples of cellophane, PET, and PA 6,6 polymer films. In the second series of experiments, NFC and NCC were used to bind two identical samples of cotton, PET, and PA 6,6 fabrics. The NFC and NCC coatings on the polymer films and fabrics were prepared using the same Protocols 1–4 as in the tests of the stability of the coating in water.

The results of the peel tests with the polymer films show similar tendency as for the experiments on the coating stability in water; that is, NCC demonstrates a higher strength as compared with NFC, and the interaction with PET is the lowest among other polymers (Figure 8). However, in contrast with the experiments in water, nanocellulose interactions with the cellophane film are much stronger than with nylon. This difference in adhesion between nanocellulose materials and cellulose substrates in the dry state and in water provides evidence that the nanocellulose coating degradation in water is affected by swelling of the coating and weakening of the interfacial hydrogen bonds, while in the dry state, the intermolecular interactions remain strong.

The outcomes changed for the peel tests using fabrics instead of films made of the same polymers. NCC binding is stronger than NFC for all cases (compare Figure 9 and Figure 10). We observe that the peel strength increases in the order cotton < PET < PA 6,6 for NFC and NCC. For both types of nanocelluloses, the pretreatment with PEI and mixing with the copolymer improves the peel strength. Similar conclusions about peel strength are applied to a blended (50:50) cotton-PET fabric (Figure 11). The most surprising result is the lowest peel strength for cotton textiles. This result was in conflict with the experiment in water (Table 1) and with the peel test for polymeric films (Figure 8) when NFC and NCC coatings showed the strongest stability and adhesion to the cellulose substrates. We hypothesized an additional factor that may impact the peel strength is the structure of the fabrics or the ability of nanocellulosic materials to infiltrate the fabric structure. The latter will result in a greater contact area between the fabric and nanocellulose and the formation of mechanical interlocks between intercalated fibrillary structures.

This hypothesis was verified with the analysis of the structure and porosity of the fabrics. The warp and weft density and mean flow pore diameter (MFPD) of the fabrics are presented in Table 2. The result indicates that warp and weft density for cotton fabrics are substantially higher than those for other samples. While warp densities between polyester, nylon, and cotton/polyester are similar, the weft density of nylon is the lowest among the samples. Fabrics with higher weft density are less permeable for functional additives [56]. This structural property explains the highest peel strength for nylon and the lowest for the cotton fabrics.

The mean flow pore diameter of the cotton and nylon fabrics also shows the highest values with broader pore size distribution in nylon fabric. This is another factor correlating with the uptake of NFC and NCC into the fabrics. Polyester has the lowest MFPD of 18 µm, but the weft density of 17 yarns/cm makes the fabric less dense for the infiltration of nanocellulose hydrogels into the fabrics, contributing to the higher peel strength as compared with the cotton fabric.

For the cotton fabric, the pore size ranges from 10 to 128 µm, while for the nylon fabric, the range is very broad, and the pore size reaches ∼200 µm. The cotton/polyester blend pore size distribution shows that the majority of the pores are less than ∼100 µm, and for the PET fabric, the pore size range is between ∼5 and ∼50 µm. The pore size distributions of the cotton, polyester, nylon, and cotton/polyester blended fabrics are shown in Appendix A. The peel test results correlate with pore size distribution as the highest peel strength is observed for the nylon fabrics with a skew distribution towards large size pores.

### 2.4. Covalent Cross-Linking of Nanocellulose Coatings

An alternative approach to stabilize nanocellulose coatings is the cross-linking of NFC and NCC particles. Cross-linking of cotton fabrics is a widely used method to fabricate wrinkle resistance cotton products. Polycarboxylic acids for cross-linking of cotton cellulose were first introduced in the 1960s. The cross-linking can be catalyzed with sodium hypophosphite NaH_2_PO_2_ [57,58,59,60,61,62]. Polycarboxylic acids, for example, maleic acid (MA), in the presence of sodium hypophosphite, form ester bonds with cellulose hydroxyls at 160–180 °C [63,64]. The formation of the cross-links is confirmed by the appearance of the ester carbonyl band at 1720 and 1718 cm^−1^ (Appendix A).

The results of the peel tests for the cross-linked NFC and NCC coatings on the cotton fabric are shown in Figure 12. For both nanocellulose materials, the cross-linked system is stronger as compared with the reference non-cross-linked materials. However, the results are comparable to those obtained with P(GMA-OEGMA) copolymer. The results with cross-linked nanocellulose materials on the cotton fabric are compared to those on the cellophane film in Figure 12. This comparison reveals the synergistic effect of cross-linking and infiltration of nanocellulosic materials into the fabric structure. Smaller NCC particles infiltrate into the dense cotton fabric structure more efficiently as compared with NFC. This infiltration results in an increased adhesive interface. At the same time, the infiltration underlines the contribution of the mechanical interlocking enforced by the cross-linking of the cellulosic materials.

## 3. Materials and Methods

### 3.1. Materials

Nanofibrillated cellulose hydrogel (2%) was prepared as previously reported using a mechanical homogenization method [37,38]. Nanocrystalline cellulose hydrogel (11.9%) was purchased from the Process Development Center, University of Maine. m-Cresol, lithium chloride, N,N-dimethylacetamide, chlorobenzene, phenol, polyethylenimine (PEI) (number average molecular mass Mn = 60 kg/mol), maleic acid (MA), sodium hypophosphite, glycidyl methacrylate (GMA, 97%), azoisobutyronitrile (AIBN), oligo (ethylene glycol) methyl ether methacrylate (OEGMA, Mn = 950 g/mol, stabilized with inhibitors), and inhibitor removers (kits for removing hydroquinone and monomethyl ether hydroquinone (MEHQ) and tert-butylcatechol (BHT)) were purchased from Sigma-Aldrich (St. Louis, MO, USA).

Silicon wafers (100 crystal plane) with a native oxide layer were purchased from University Wafer (South Boston, MA, USA). A polyethylene terephthalate (PET) film (0.50 mm thick) and a nylon 6,6 (PA 6,6) film (0.5 mm thick) were purchased from GoodFellow, Coraopolis, PA, USA. A cellophane film (regenerated cellulose) 0.03 mm thick was provided by Thermetrics. Cotton (100%, plain weave), nylon 6,6 (100%, spun, plain weave), cotton/polyester, PET (50%/50%, plain weave), polyester, and PET (100%, spun, plain weave) fabrics were purchased from Testafabrics, Inc., West Pittston, PA, USA.

### 3.2. Synthesis of P(GMA-OEGMA) Copolymer

P(GMA-OEGMA), a random copolymer of glycidyl methacrylate (GMA) and oligo(ethylene glycol) methacrylate (OEGMA), M_w_ = 2000 kg/mol, was synthesized by solution free-radical polymerization [65,66]. The inhibitor MEHQ and BHT removers were added to purify GMA and OEGMA for 45 min each. After filtration and purging with nitrogen for 45 min, the monomer solution (0.5 M) at GMA/OEGMA + 1:4 in MEK was used for polymerization initiated by 0.01 M AIBN at 50 °C for 1.5 h. The copolymer was extracted and purified by repetitive precipitation (three times) in diethyl ether. The copolymer was stored in a MEK solution in the absence of light. According to an NMR analysis (Bruker AVANCE-300, Billerica, MA, USA), the copolymer composition is 66 mol% (93 wt%) OEGMA and 34 mol% GMA. The copolymer is characterized by a glass transition temperature of −50 °C and a melting temperature of 35 °C (DSC2920, TA Instruments, New Castle, DE, USA). The copolymer is soluble in water.

### 3.3. Preparation of Polymer Substrates for Nanocellulose Deposition

Si-wafers were cut into square pieces (1 cm × 1 cm) and then cleaned in a solution of 28% NH_4_OH/30% H_2_O_2_/H_2_O (1:1:1) at 65 °C for 1 h. They were rinsed with deionized water (DI) water and dried under a flux of argon gas. The cleaned wafers were dipped into a solution of PEI (1%) for 15 min and rinsed with DI water and dried with argon gas. The resulting modified silicon wafers were stored at room temperature in a clean desiccator.

PET was dissolved in a solution of phenol-chlorobenzene (1:1) in a boiling water bath (100 °C). Once fully dissolved, 20 μL of the solution was spin-coated (3000 rpm for 20 s) on the Si-wafers. The substrates were transferred to an oven heated at 180 °C and annealed for 24 h to remove the residual solvent.

PA 6,6 was dissolved in m-cresol at 100 °C by stirring for several hours. The nylon films were prepared by spin-coating (3000 rpm for 20 s) 20 μL of the solution on the Si-wafers. The substrates were immediately transferred to an oven and dried at 180 °C for 24 h.

Cellulose films were prepared by heating of a cellulose powder in a solution of lithium chloride (LiCl, 1–3%) and N,N-dimethylacetamide (DMAc, 3–9%) to 150 °C, and then allowed the solution to cool slowly to room temperature [67,68]. Afterward, 20 μL of the cellulose solution was spin-coated (3000 rpm for 10 s) onto the cleaned Si-wafers. The samples were submerged in DI water for 20 min to remove the remaining LiCl; they were dried with argon gas and heated for 20 min at 180 °C to allow for the evaporation of any residual solvent. The samples were stored at room temperature.

### 3.4. Nanocellulose Coatings on Si-Wafers and Polymer-Coated Si-Wafers

NFC and NCC films were prepared by spin-coating (3000 rpm for 30 s) 20 μL of the diluted NFC and NCC hydrogels (1%) onto the substrates (Si-wafers and cellulose, PET, and PA 6,6 coated Si-wafers) and annealed at 120 °C for 24 h.

Nanocellulose coatings were prepared on the surface of CL, PET, and PA 6,6 coated Si-wafers after pretreatment of the substrates with PEI or P(GMA-OEGMA). These samples were labeled as PEI-NFC and PEI-NCC, respectively. These samples were prepared by submerging of the polymer-coated Si-wafers into a 1% PEI solution for 30 min. Then, the samples were rinsed with DI water and dried with argon gas. NFC and NCC solutions were spin coated (3000 rpm 30 s). The samples were annealed at 120 °C for 1 h.

Alternatively, NFC (0.1%) and NCC (1%) were mixed with a 1% PEI or P(GMA-OEGMA) solution (20:1 by volume) and stirred for 1 h, for nanocellulose-PEI and nanocellulose- P(GMA-OEGMA) blends, respectively. Then, 20 μL of the PEI-modified or P(GMA-OEGMA)-modified hydrogels was spin-coated onto the polymer-coated Si-wafers and dried at 120 °C for 1 h. These samples were labeled PEI+NFC, PEI+NCC, CP+NFC, and CP+NCC, for NFC and NCC mixed coatings, respectively.

### 3.5. Characterization of Coatings

Scanning electron microscopy (SEM) imaging was carried out using an FEI Teneo (FEI Co., Hillsboro, OR, USA), a field emission scanning electron microscope. Atomic force microscopy (AFM) images were obtained using a Bruker Multimode Nanoscope instrument (Bruker, Billerica, MA, USA) with the ScanAssyst-Air probe (Bruker) spring constant 0.4 N/n, a silicon oxide tip). All the measurements were performed under ambient conditions at room temperature and at a relative humidity (RH) of 50–55%. All AFM data analysis and data processing were done with the NanoScope Analysis software version 1.40 (Bruker).

The thickness of the films at three different locations for each sample after the deposition of each layer on the substrates was measured by a single wavelength imaging ellipsometer ep4sw (Accurion, Göttingen, Germany) with a fixed angle of incidence of 70°. Ellipsometry thickness maps were generated using the Accurion software package, DataStudio, for selected samples to verify the uniformity of the coatings and justify measurements of the samples series using the three-location-approach. An attenuated total reflection fourier transform infrared spectroscopy (ATR-FTIR, ThermoElectron Nicolet 6700) was used to collect the infrared spectra. AFM was used as an alternative method for coating thickness by scratching the coating with a steel needle and measuring the profile of the scratch (Appendix A).

The spectra were presented using absorbance mode (−logR/R_o_). The resolution for all the infrared spectra was 4 cm^−1^, 120 scans for each spectrum.

### 3.6. Structural Characterization

A porometer (Porous Materials Inc., Ithaca, NY, USA) was used to measure the fabric porosity. A capillary flow porometer (CFP) was used to evaluate an average pore diameter, pore size distribution, and mean flow pore size by assessing the relationship between pressure and gas flow rate [69]. The fabrics were cut into circular pieces of 25 mm in the diameter, soaked into Galwick wetting liquid with the surface tension of 16 Dyne/cm, placed, and sealed into the sample holder for the measurements.

### 3.7. Cross-Linking of Nanocellulose Coatings

The cross-linking was conducted as published elsewhere [63,64], after drying the nanocellulose coatings were treated with 6%MA and 4% NaH_2_PO_2_ solutions and annealed at 185 °C for 2 min.

### 3.8. T-Peel Tests

The tests were performed according to the method described in the ASTM D1876-08 standard with five tested samples for each material. Samples of the fabrics were cut into 50 mm × 152 mm stripes along the warp direction. The test panel (Appendix A) consisted of two fabrics stripes bonded together with 5 g of 2% NFC or NCC hydrogels along 127 mm of their length. A 2 kg load was applied to the top of the test panels while drying at 85 °C. Then, the specimens were placed in a conditioning chamber (Caron^®^) for 7 days at a relative humidity of 50 ± 2% at 23 ± 1 °C. Finally, we cut the bonded panels into 25 mm wide test specimens with a sharp cutter. During the tests, the peeling force was recorded at a constant head speed of 254 mm/min. The same method for the sample preparation was used for PEI and P(GMA-OEGMA) blended NFC and NCC hydrogels, respectively, as well as for the samples with cross-linked nanocellulose coatings. In the latter case, the samples were annealed at 185 °C for 2 min. Using ICPeel software [70], we estimated that the bending energy contributions is 15%, which is comparable to the peel test experimental error. Variations in nanocellulose coating thickness in a range of 1–5 µm have a very low effect on the contribution of the bending energy.

## 4. Conclusions

Wet and dry tests of nanocellulose coatings on the surface of cellulose, PET, and nylon (PA 6,6) polymer films revealed that the coatings have the highest adhesion to the nylon, cellulose, and cellophane surfaces, while adhesion is the lowest for the coatings on PET. The coatings stability is improved using treatment with a polycation polymer PEI and a reactive P(GMA-OEGMA) copolymer capable of forming a cross-linked network. In the latter case, the highest coating adhesion and stability were observed. Alternatively, the coating is reinforced by the cross-linking of nanocellulose with polycarboxylic acids. NCC coatings demonstrate higher adhesion to all substrates than NFC coatings. The experiments with cotton, PET, and PA 6,6 fabrics revealed that the fabric structure is an additional important factor for the stability and adhesion of the nanocellulose coatings. The lower density of the textile and higher porosity is beneficial for stronger adhesion of the coatings. Hydrogen bonding, swelling in water, physical, covalent cross-linking, overall contact area, and porosity of the substrate, which provide intercalation of the nanocellulose particles into the fabric structure, are all characteristics that contribute to the nanocellulose coating adhesion and stability.

## Figures and Tables

**Figure 1 molecules-25-03238-f001:**
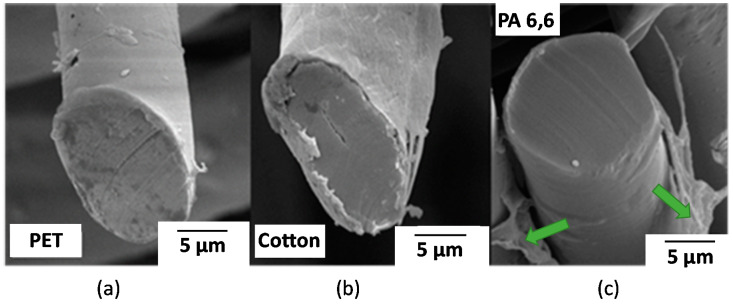
Scanning electron microscopy (SEM) images of (**a**) poly(ethylene terephthalate) (PET), (**b**) cotton, and (**c**) nylon 6,6 (PA 6,6) single fibers coated with nanofibrillated cellulose (NFC). Arrows point to the peeled fraction of the NFC coating.

**Figure 2 molecules-25-03238-f002:**
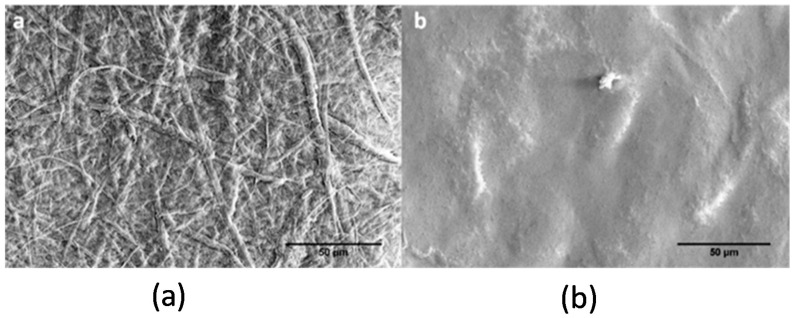
SEM images of the morphology of (**a**) NFC and (**b**) nanocrystalline cellulose (NCC) spin-coated on the Si-wafers.

**Figure 3 molecules-25-03238-f003:**
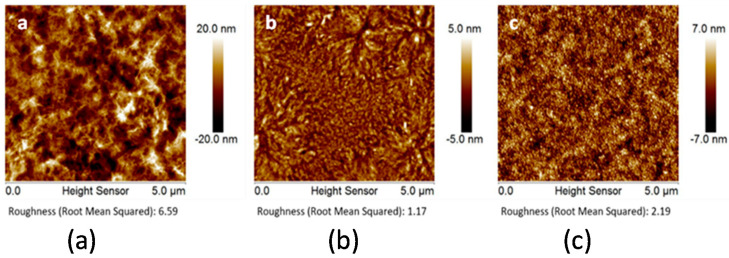
Representative atomic force microscopy (AFM) images of about 50–180 nm thick spin-coated films of (**a**) cellulose (CL), root mean square (RMS) roughness is 6.59 nm; (**b**) nylon, RMS roughness is 1.17 nm; and (**c**) PET, RMS roughness is 2.19 nm on the surface of the Si-wafers.

**Figure 4 molecules-25-03238-f004:**
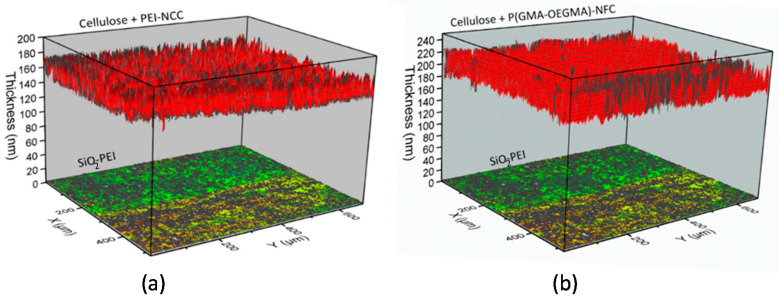
Representative examples of 3D-plots of layer-by-layer ellipsometric mapping of nanocellulose films on the polymer-coated Si-wafers constituted of the layers: (**a**) native SiO_2_ (1 nm), poly(ethylene imine) (PEI) (1 nm), cellulose film (150 nm), PEI (1 nm), and NCC (20 nm); (**b**) native SiO_2_ (1 nm), PEI (1 nm), CL (180 nm), PEI (1 nm), and NFC-copolymer of glycidyl methacrylate (GMA) and oligo(ethylene glycol) methacrylate (OEGMA) (P(GMA-OEGMA)) mixture (40 nm).

**Figure 5 molecules-25-03238-f005:**
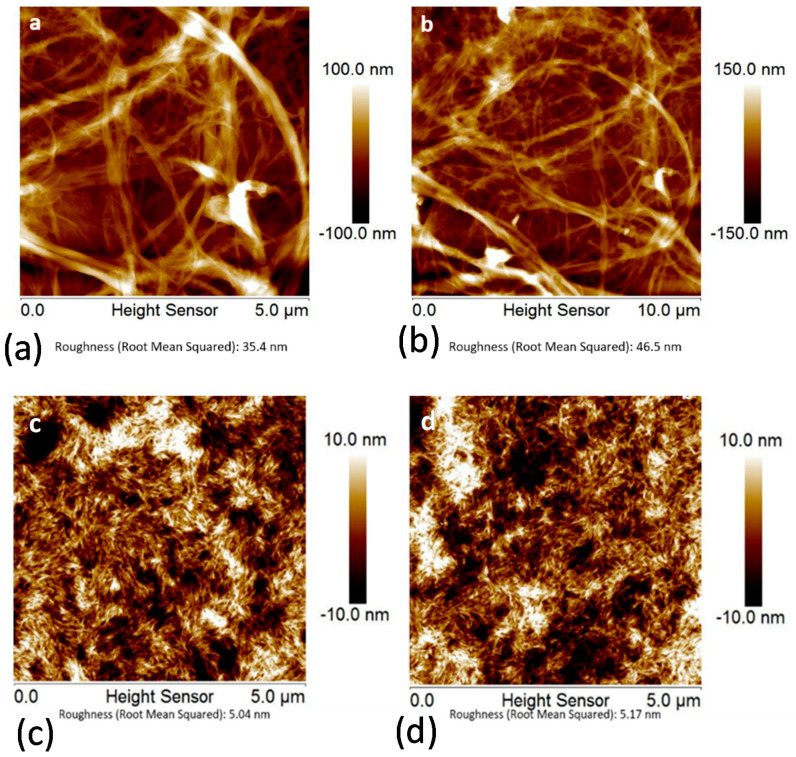
AFM topography images of nanocellulose coatings (**a**,**b**) PEI-NFC and (**c**,**d**) NCC+PEI on (**a**,**c**) PA 6,6 as-deposited and (**b**,**d**) after rinsing in water.

**Figure 6 molecules-25-03238-f006:**
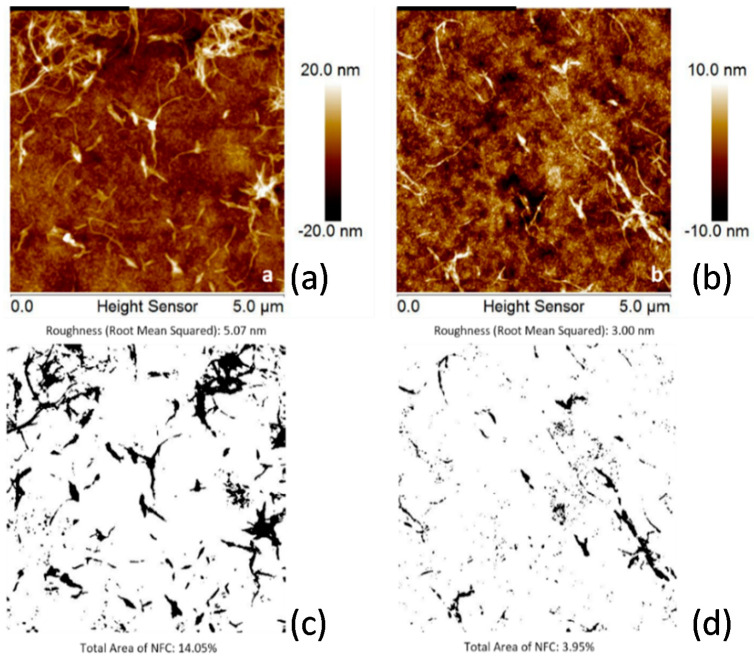
AFM images of NFC coatings on PET (**a**,**c**) as-deposited and (**b**,**d**) after rinsing in water: (**a**,**b**) topography images and (**c**,**d**) flooding analysis images showing the surface coverage by NFC particles (dark areas).

**Figure 7 molecules-25-03238-f007:**
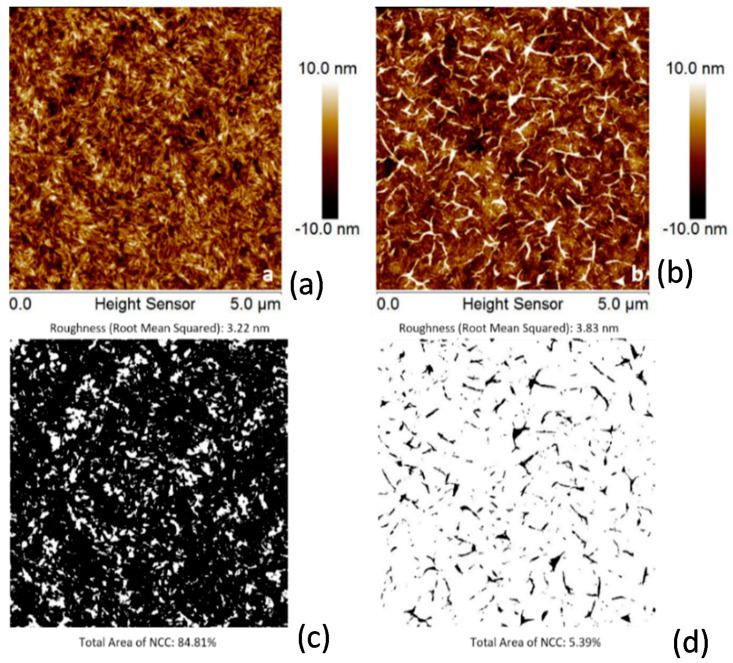
AFM images NCC coatings on PET as-deposited (**a**,**c**) and after rinsing in water (**b**,**d**): topography images (**a**,**b**) and flooding analysis images showing the surface coverage by NCC particles (**c**,**d**).

**Figure 8 molecules-25-03238-f008:**
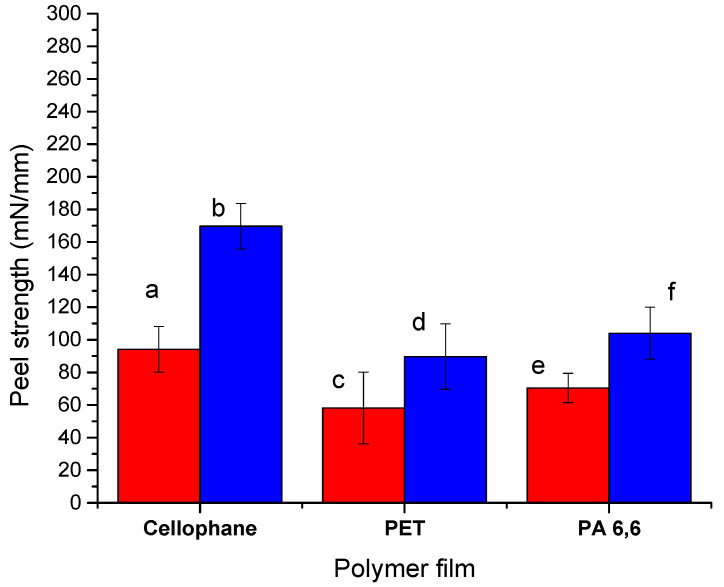
Peel strength for polymer films made of cellophane, PET, and PA 6,6 adhered with (**a**,**c**,**e**) NFC and (**b**,**d**,**f**) NCC.

**Figure 9 molecules-25-03238-f009:**
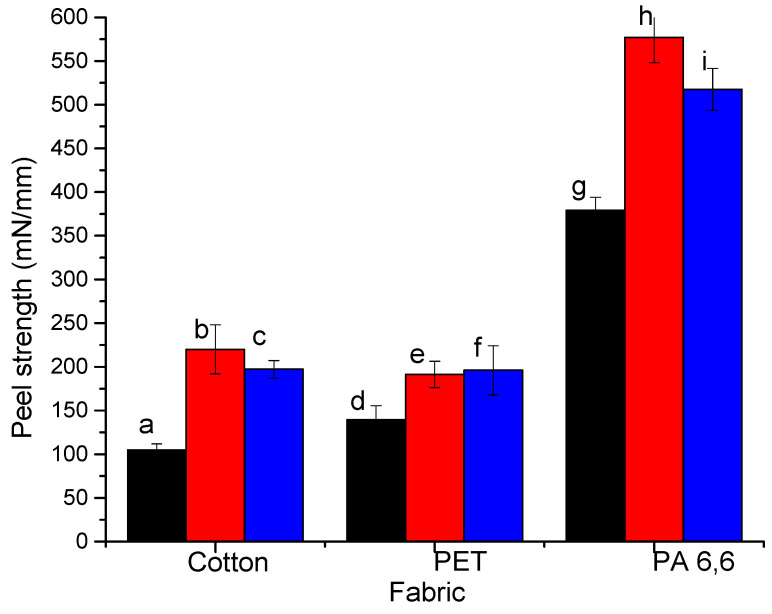
Peel strength for cotton, PET, and PA 6,6 fabrics adhered using NFC: (**a**,**d**,**g**) NFC with no additives, (**b**,**e**,**h**) NFC+PEI, and (**c**,**f**,**i**) NFC+P(GMA-OEGMA) mixtures.

**Figure 10 molecules-25-03238-f010:**
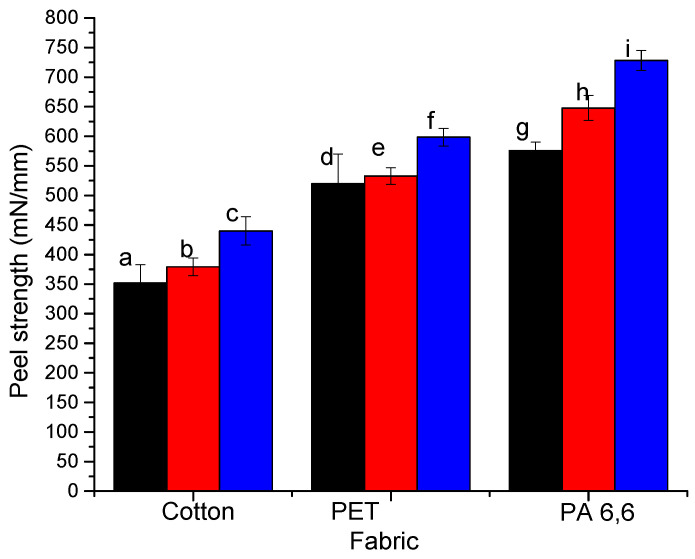
Peel strength for cotton, PET, and PA 6,6 fabrics adhered using NCC: (**a**,**d**,**g**) NCC with no additives, (**b**,**e**,**h**) NCC+PEI, and (**c**,**f**,**i**) NCC+P(GMA-OEGMA) mixtures.

**Figure 11 molecules-25-03238-f011:**
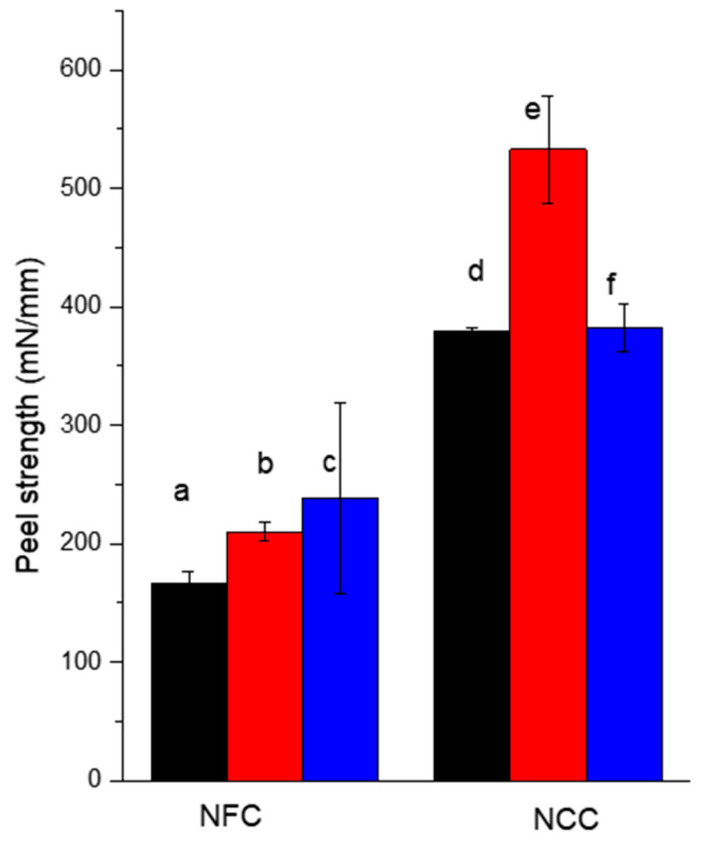
Peel strength for cotton-PET (50:50) fabric blend adhered with (**a**,**d**,**c**) NFC and (**d**–**f**) NFC: (**a**,**d**) with no additives, (**b**) NFC+PEI, (**e**) NCC+PEI, (**c**) NFC+P(GMA-OEGMA), and (**f**) NCC+P(GMA-OEGMA) mixtures.

**Figure 12 molecules-25-03238-f012:**
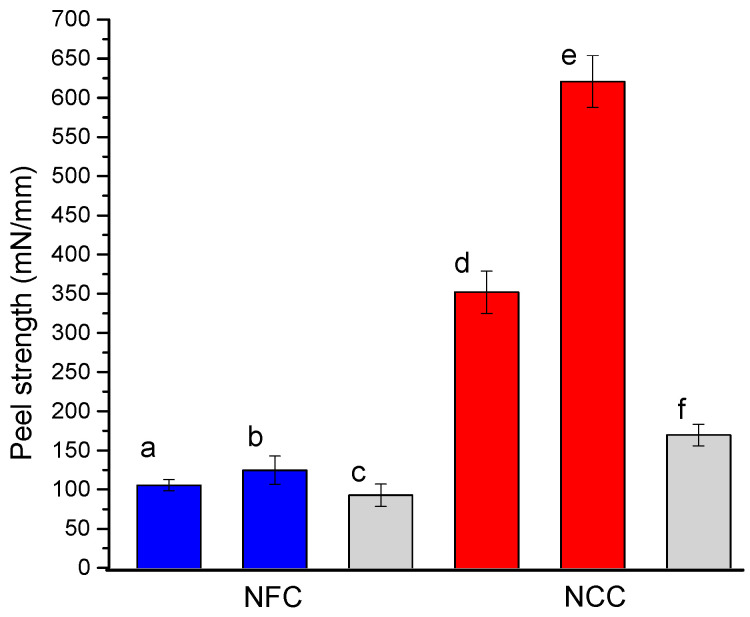
Peel strength for cotton textiles adhered with (**a**,**b**) NFC and (**d**,**e**) NCC with (**a**,**d**) no cross-linking and (**b**,**e**) with covalent cross-linking using MA compared with (**c**) NFC and (**f**) NCC adhered cellophane films.

**Table 1 molecules-25-03238-t001:** Changes in thickness (surface coverage) of nanofibrillated cellulose (NFC) and nanocrystalline cellulose (NCC) coatings on the surfaces of cellulose (CL), poly(ethylene terephthalate) (PET), and nylon 6,6 (PA 6,6) after rinsing in water. PEI, poly(ethylene imine); P(GMA-OEGMA), copolymer of glycidyl methacrylate (GMA) and oligo(ethylene glycol) methacrylate (OEGMA).

Sample	Film Thickness, H, and A fraction of Washed-Out Coating from Different Substrates, F (10% Error)					
	CL		PET		PA 6,6	
	H, nm	F, %	H, nm	F, %	H, nm	F, %
NFC	50	68	non-uniform	71	non-uniform	40
PEI-NFC	46	22	42	18	-	-
NFC+PEI	20	70	77	81	53	37
NFC+P(GMA-OEGMA)	33	77	-	-	401	25
NCC	21	5	non-uniform	93	non-uniform	93
PEI-NCC	89	53	113	60	-	-
NCC+PEI	40	25	77	12	65	77
NCC+P(GMA-OEGMA)	8	11	80	1	133	2

**Table 2 molecules-25-03238-t002:** Structural characteristics of the fabrics. MFPD, mean flow pore diameter.

Samples	Warp Density, Yarns/cm	Weft Density, Yarns/cm	MFPD, µm
**Cotton**	33	29	67
**Polyester**	22	17	18
**Nylon**	20	11	64
**Cotton/Polyester (50%/50%)**	21	19	40

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
