# Peer review of "Adhesion and Stability of Nanocellulose Coatings on Flat Polymer Films and Textiles"

_molecules, 2020, doi:10.3390/molecules25143238_

Round 1
Reviewer 1 Report
I was not able to locate the supplementary data and would find this interesting to see.
Nanocellulose is an interesting material and more knowledge is important on how it can be used. Lacking is in the introduction why these coatings are important to look at and what is the suggested use. The authors suggest nanocellulose as environmental friendly - and suggest quite a chemical method.
Line 107 - mistake with a missing reference.
The peel off from the coated fibers - are this an important and useful effect? Or just useful for the measuring?
Fig 3 - roughness more decimal in the figure than in the legend.
The authors show how much can be washed off and how much is left behind on the surface. Are they any meassuring how the attachment is?
Fig 5 - legend for ab write PEI-NFC - should this be NFC+PEI?
Reviewer 2 Report
The manuscript reports a series of preparations and testing of nanocellulose coatings on polymer films and textiles in terms of adhesion and mechanical stability. The subject is interesting and will likely have a broader readership. It might be publishable after the following issues and questions are addressed.
1) the pretreatment of polymers with UV Ozone or plasma activation is a common approach to improve coating adhesion. It is surprising that nothing about it was mentioned in this paper. The relevant literature needs to be described and discussed in the Introduction - see this paper for related information: Cellulose (2017) 24:1877–1888. It is also necessary to provide a justification of using chemical additives instead of the activation pretreatment methods.
2) polymer film and fabrics are flexible and often subject to bending deformation. How could the polymer bending affect the adhesion and stability of the nanocellulose coatings?
3) Table 1 refers to the polymer films but the descriptions in the paragraph right beneath the table refers to the fabrics. Please clarify the inconsistence.
4) the film thickness varied a lot among the different samples in Table 1. what are the effects of the different treatments on the coating thickness?
5) Could the infiltration and crosslinking of the nanocellulose materials into the fabric change the stiffness or flexibility of the fabrics? The stiffness change is also related to the measured peel strength.
Reviewer 3 Report
The content of the article is definitely of interest for the commmunity working on nancellulose. The manuscript needs to be improved in some parts before being published:
- pg2, line 70: "strongly hydrogen-bonded nanocellulose materials swell in an aqueous environment." This is a questionable claim as a lot of literature reports opposite results (see eg https://www.sciencedirect.com/science/article/pii/S0144861718314085?via%3Dihub).
- pg 3 line 107: missing reference
- pg 8, fig 7c. AMF "flooding analysis" is fine to quantify the amount of material removed/still lattached but only if the underlying substrate is clearly visible. This is not the case of Fig7c where there is no clear evidence of a flat substrate. The CNC are too dense to assume that the lowest point is the substrate level. Why authors do not complement the afm analisys with ellipsometry?
- results of table1 are definitely confuse: measurements must be repeated from at least few times to get an idea of the reproduciblity of the adhesion/detachment. Furthermore, since AFM measure very small area, a statistical analysis over some images have to be done. Only after data about the reproducibility of the experiment are presented, the authors can draw conclusion about the best/worst adhesion behavior.
- pg 10, line 249: it is not clear which protocol have been used for the subsequent experiments: what does it means pei-treated? is it the NCCmixed with PEI or is the fabric wetted by pei before covering it with NC?
- pg 12, from line 289: the hypothesis is rather speculative. the authors are comparing lenghtscales covering 7 orders of magnitudes (from cm to nm). The interrelations between warp/weft(cm) with MFPD (um) and NC(nm) on the adeshion cannot be argue from these simple experiments. At least the warp/weft amd MFPD parameters should be varied independently to demosntrate a a causal correlation of the data
- pg 15, line 399: how do the authors construct a 2D map from 3 point measurements (since graphs reported in fig1 are made of thousands of points)?
- pg 15, sec3.8: please add the number of tests performed for each material
Round 2
Reviewer 2 Report
The quality of the revised manuscript has much improved. I am satisfied with the revision and responses.
Reviewer 3 Report
just one small comment. I noticed that RMS values are iven with two decimal nnumbers (e.g. 2.19 nm) this is not realistic (AFM electronic noise is of the order of 0.02nm, plus the noise during the measure). A single decimal is enough.